# Unveiling the bosonic nature of an ultrashort few-electron pulse

Gregoire Roussely [1], Everton Arrighi[1], Giorgos Georgiou[1,2], Shintaro Takada [1,3], Martin Schalk[1], Matias Urdampilleta[1], Arne Ludwig[4], Andreas D. Wieck [4], Pacome Armagnat[5], Thomas Kloss[5], Xavier Waintal[5], Tristan Meunier[1] & Christopher Bäuerle [1]

Quantum dynamics is very sensitive to dimensionality. While two-dimensional electronic systems form Fermi liquids, one-dimensional systems—Tomonaga–Luttinger liquids—are described by purely bosonic excitations, even though they are initially made of fermions. With the advent of coherent single-electron sources, the quantum dynamics of such a liquid is now accessible at the single-electron level. Here, we report on time-of-flight measurements of ultrashort few-electron charge pulses injected into a quasi one-dimensional quantum conductor. By changing the confinement potential we can tune the system from the one-dimensional Tomonaga–Luttinger liquid limit to the multi-channel Fermi liquid and show that the plasmon velocity can be varied over almost an order of magnitude. These results are in quantitative agreement with a parameter-free theory and demonstrate a powerful probe for directly investigating real-time dynamics of fractionalisation phenomena in low-dimensional conductors.

[1] Univ. Grenoble Alpes, CNRS, Grenoble INP, Institut Néel, 38000 Grenoble, France. [2] Univ. Savoie Mont-Blanc, CNRS, IMEP-LAHC, 73370 Le Bourget du Lac, France. [3] National Institute of Advanced Industrial Science and Technology (AIST), National Metrology Institute of Japan (NMIJ), Tsukuba, Ibaraki 305-8563, Japan. [4] Lehrstuhl für Angewandte Festkörperphysik, Ruhr-Universität Bochum, Universitätsstrasse 150, 44780 Bochum, Germany. [5] Univ. Grenoble Alpes, CEA, INAC-Pheliqs, 38000 Grenoble, France. These authors contributed equally: Gregoire Roussely, Everton Arrighi. Correspondence and requests for materials should be addressed to C.B. (email: christopher.bauerle@neel.cnrs.fr)

A fundamental difference between bosons and fermions is that the former can be described at the classical macroscopic level while the latter cannot. In particular, in an ultrafast quantum nanoelectronics setup, the experimentalist controls the—bosonic—electromagnetic degrees of the system and aims at injecting a single—fermionic—coherent electron in the system. This interplay between bosonic and fermionic statistics is a central feature in one-dimensional quantum systems as it provides a unique playground for the study of interaction effects[1,2].

The reduced dimensionality influences the interaction between particles and can lead to fascinating phenomena such as spin–charge separation[3], charge fractionalisation[4] or Wigner crystallisation[5]. The low-energy collective bosonic excitations consist of charge and spin density waves that propagate at two different velocities. While the spin density is unaffected by the Coulomb interaction and propagates at Fermi velocity $v_F$, the charge density is strongly renormalised by the interactions and propagates with the plasmon velocity $v_P$, which is usually much faster than the Fermi velocity. Spin–charge separation has been experimentally probed in momentum resolved tunnelling experiments between two quantum wires[3] as well as tunnelling from a quantum wire into a two-dimensional electron gas[6]. In addition to spin–charge separation, charge fractionalisation occurs in one-dimensional systems[7–10]. Injecting an electron into a one-dimensional system with momentum conservation, the charge decomposes into right and left moving charge excitations, as demonstrated in ref. [4]. Charge fractionalisation also occurs in a system of two coupled Tomonaga–Luttinger liquids. There, an electronic excitation present in one of the two channels fractionalises into a fast charge mode and a slow neutral mode, which are the eigenmodes of the coupled system[11]. This charge fractionalisation has been recently observed in a chiral two-channel Tomonaga–Luttinger liquid in the integer quantum Hall regime[12–16].

Here, we study the most general case where the system can be tuned continuously from a clean one-channel Tomonaga–Luttinger liquid to a multi-channel Fermi liquid in a non-chiral system. We use time-resolved measurement techniques[17,18] to determine the time of flight[19–21] of a single-electron voltage pulse and extract the collective charge excitation velocity. Our detailed modelling of the electrostatics of the sample allows us to construct and understand the excitations of the system in a parameter-free theory. We show that our self-consistent calculations capture well the results of the measurements, validating the construction of the bosonic collective modes from the fermionic degrees of freedom.

## Results

**Measurement principle**. We tailor a 70 µm long quasi one-dimensional wire into a two-dimensional electron gas using metallic surface gates as shown in Fig. 1a. A pump-probe technique has been implemented to measure in a time-resolved manner the shape as well as the propagation speed of the electron pulse. We apply an ultrashort voltage pulse (≈70 ps) to the left ohmic contact to generate the few-electron pulse. The pulse injection is repeated at a frequency of 600 MHz and the resulting DC current is measured at the right ohmic contact. Three quantum point contacts (QPCs) are placed along the quantum wire to measure the arrival time of the charge pulse at different positions. Simultaneously, another ultrashort voltage pulse is sent to one of the three QPCs which allows opening and closing the QPC on a timescale much faster than the width of the few-electron pulse (see Methods section). By changing the time delay between launching the electron pulse and the on–off switching of

the QPC, we can reconstruct the actual shape of the few-electron pulse[19,20].

**Time-of-flight measurements**. A typical time-resolved measurement is shown in Fig. 1b. We observe a few-electron pulse of Gaussian shape with a full width at half maximum (FWHM) of ≈70 ps. Measurements of the time of flight $\tau_F$ at different positions (Fig. 1b) allows us to determine its propagation speed, which we find to be independent of the number of electrons contained in the electron pulse (Fig. 1d). By changing the voltage on the side gates $V_{SG}$ it is possible to modify the propagation speed by almost an order of magnitude. As the confinement is made stronger, the arrival time of the electron pulse at the detection QPC is shifted to longer times, as seen in Fig. 2a. This, as it will be demonstrated further on, is an indication of a slower propagation speed and it is in stark contrast to standard DC measurements. Indeed, in DC the Coulomb interaction is screened by the Fermi sea and the electrons travel at the Fermi velocity, as shown by magnetic focussing experiments[22]. The situation is very different when creating a local perturbation of the charge density. Applying a very short charge pulse results in an excess charge density created locally. Due to the generated electric field, the excess charge is displaced very rapidly at the surface of the Fermi sea giving rise to a collective excitation, a plasmon[23].

**Effect of Coulomb interaction on the propagation velocity**. In one dimension, an interacting wire is described by Tomonaga–Luttinger plasmons of bosonic character[1]. The problem of generalising the bosonization construction to a system containing an arbitrary number of conduction channels, $N$, in the presence of Coulomb interactions has been treated theoretically by Matveev and Glazman[24]. The effect of the Coulomb potential is to couple the individual channels of the quantum wire, thus resulting in a collective behaviour that in turn affects strongly the propagation velocity of the excitations. For a quantum wire containing $N$ conduction channels, Coulomb interaction leads to charge fractionalisation into $N$ charge modes with renormalised propagation velocity and $N$ spin modes (c.f. Supplementary Note 4). To distinguish between single-particle states and collective modes, we will use throughout this study the term channel whenever referring to single-particle states and mode when referring to collective modes. As the spin modes do not carry any charge, their speed is not affected by the Coulomb interaction. For our experiment we can neglect them since voltage pulses do not excite spin modes in the quantum conductor. The $N$ charge modes, on the other hand, are affected by the Coulomb interaction in the following way: $N-1$ charge modes—the slow modes—are weakly affected and propagate with a speed close to $v_F$, while one mode—the fast mode—usually referred to as the plasmon mode is renormalised via all the other modes and propagates with a velocity much faster than $v_F$.

Here, we have derived the theory[24] from first principles in order to obtain a quantitative—parameter-free—comparison with the measurements. Our calculations proceed in three steps. First, we solve the self-consistent electrostatics-quantum mechanics problem to obtain the effective potential seen by the electrons as shown in Fig. 1c. Second, we compute the effective propagating channels and their interaction matrix. Third, we compute the mode velocities as arising from bosonisation theory (c.f. Supplementary Note 4). The obtained theoretical data for the fast mode—the plasmon mode—(without any adjustable parameters) are displayed by the blue curve in Fig. 2b.

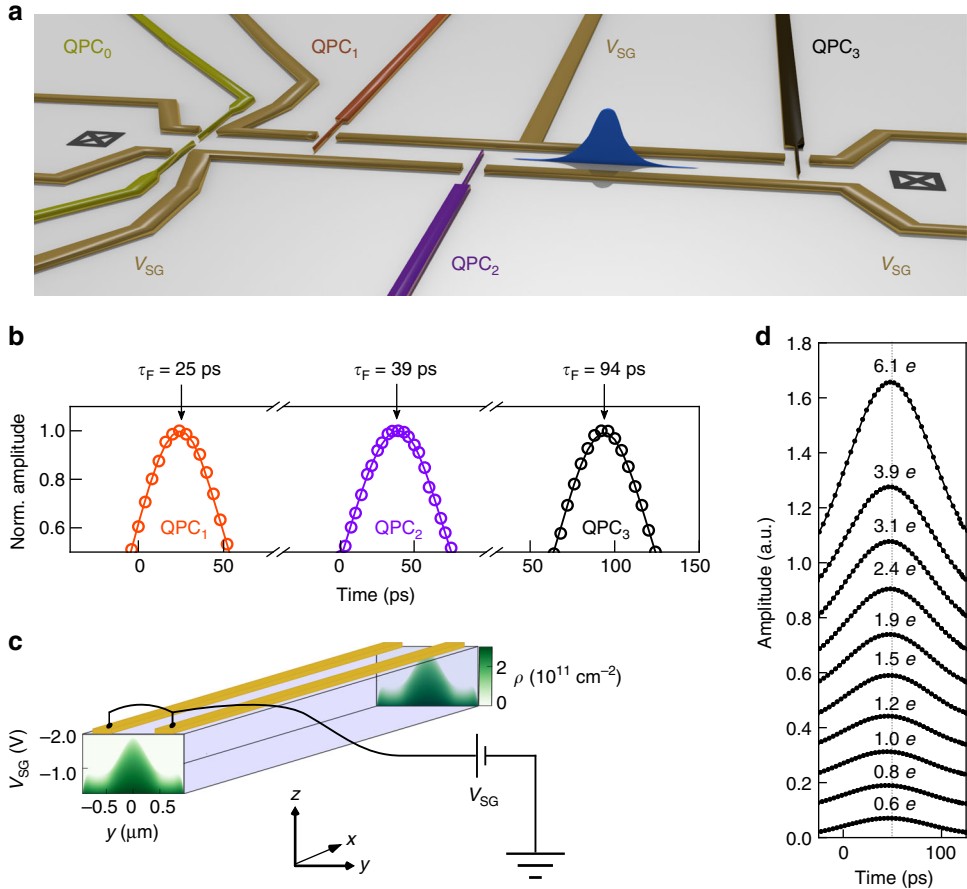

**Fig. 1** Device and time-of-flight measurements. **a** Schematic of the quantum device. A few-electron pulse is launched at the left ohmic contact (black crossed box) by applying a very short (≈70 ps) voltage pulse. Three QPCs, denoted as $QPC_1$, $QPC_2$ and $QPC_3$, are placed along the quantum device at a distance of 15, 30 and 70 μm from the left ohmic contact. Each of these three QPCs is connected to a large bandwidth (40 GHz) bias tee and operated as an ultrafast switch. Time-resolved detection of current is done at the right ohmic contact. $QPC_0$, placed a distance of 6 μm from the left ohmic contact, is used as a channel selection. **b** Time-resolved measurements of an electron pulse at the three different QPC positions. **c** Illustration of the sample geometry used for the self-consistent calculations. The quasi one-dimensional quantum wire is defined by the two long electrostatic gates at potential $V_{SG}$. The coloured images, one at the beginning of the wire and another one at the end, are cross-sections of the electron density profile along the y-axis as a function of the gate voltage. **d** Time-resolved measurements of an electron pulse at $QPC_3$ for different excitation amplitudes. The amount of electrons contained in the electron pulse is varied between 0.6 e and 6.1 e

**Channel selection**. By gradually reducing the number of channels of the quantum wire to one, we enter the Tomonaga–Luttinger liquid regime[1]. However, due to the strong confinement potential and its long length, the quantum wire is not very homogeneous and the pulse becomes distorted. It is therefore not possible to realise a clean one-channel Tomonaga–Luttinger liquid[25] in the present configuration. To circumvent this limitation we have placed another quantum point contact $QPC_0$ at the entrance of the quantum wire in order to select specific channels as schematised in Fig. 3c. We set the confinement potential of the quantum wire to a situation where the wire width is relatively large ($V_{SG} = -1.0$ V; $N \approx 28$) and set the quantum point conductance to a value of $G_{QPC_0} = \frac{2e^2}{h}$.

An electron pulse is launched from the left ohmic contact into the quantum wire containing initially $N = 28$ channels. Upon propagation, this charge pulse decomposes onto the $N = 28$ eigenmodes (plasmon modes) due to Coulomb interaction. When this pulse passes through the $QPC_0$, only one channel is transmitted, as shown in Fig. 3c. After the passage, the electron pulse continues its propagation along the quantum wire containing again the same number $N$ of available channels as before the passage through the selection $QPC_0$. Assuming a non-adiabatic

passage, the charge pulse should instantaneously fractionalise into a fast plasmon mode and $N - 1$ slow modes. Very surprisingly, this is not the case. Time-resolved measurements of the charge pulse propagation through $QPC_1$, $QPC_2$ and $QPC_3$ allow us to determine the average speed of the charge pulse after passing through the selection $QPC_0$. We observe that the charge pulse is strongly slowed down after passing the channel selection $QPC_0$, as shown in Fig. 3b, d. These measurements are repeated for different confinement potentials to corroborate our findings (see red data points in Fig. 2b).

**Discussion**

As discussed above, the propagation speed of the charge pulse is strongly enhanced by the Coulomb interaction. Applying our parameter-free model we are able to determine the propagation velocity for any gate configuration. This is done for the fast charge mode in Fig. 2b (see blue continuous curve). The agreement with the experiments over the entire gate voltage region is quite remarkable. We attribute the observed discrepancy in the limit of large number of channels $N \sim 20$–$40$ to interchannel forward scattering which is not taken into account in ref.[24].

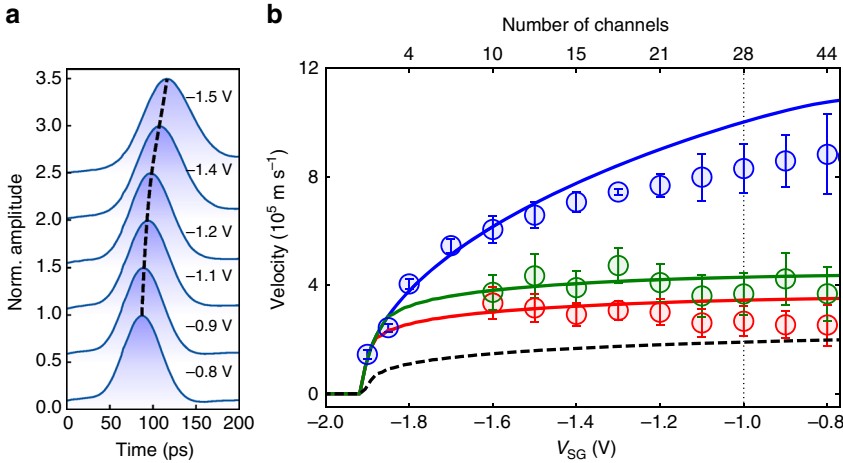

**Fig. 2** Tuning the propagation velocity. **a** Time-resolved measurements of the electron pulse for different confinement potentials at $QPC_3$ position. The curves have been offset vertically for clarity. **b** Velocity of the electron pulse as a function of the confinement potential and the corresponding number of channels. Open circles: experimental data, solid lines: parameter-free self-consistent calculation. The blue data points correspond to the situation where the channel selection $QPC_0$ is not activated. The red (green) data are obtained by setting the channel selection $QPC_0$ at a conductance value of $G = 2e^2/h$ ($G = 4e^2/h$). The red (green) solid line corresponds to the velocity of the fast plasmon mode for a one-channel (two-channel) Tomonaga–Luttinger liquid. The black dashed line corresponds to the non-interacting Fermi liquid. The vertical dotted line indicates the gate voltage at which the velocities of Fig. 3d are taken. The errors bars correspond to the velocity uncertainty and are derived from a linear fit of the QPC distance versus the respective time-of-flight (c.f. Supplementary Note 3)

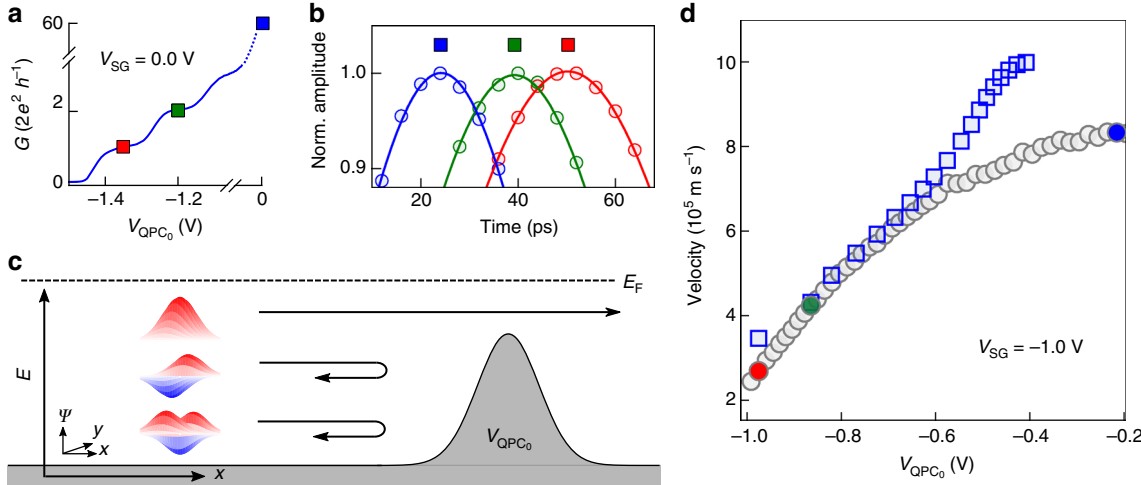

**Fig. 3** Channel selection of an electron pulse. **a** Conductance trace of the channel selection $QPC_0$ for $V_{SG} = 0.0$ V. **b** Time-resolved measurements of the electron pulse propagation detected at $QPC_1$ position, when $QPC_0$ is deactivated (blue), or set to a conductance of $G = 2e^2/h$ (red) and $G = 4e^2/h$ (green). **c** Schematic of the mode filtering experiment. The propagating voltage pulse populates all the available plasmon modes of the quantum wire (here for illustration purposes we show three) before passing the channel selection $QPC_0$. The $QPC_0$, which is set to $G = 2e^2/h$ reflects all channels except the one with the highest kinetic energy. After passing the channel selection $QPC_0$, only one single-channel plasmon mode is populated over a propagation distance of 25 μm. **d** Propagation velocity of the electron pulse as a function of the channel selection $QPC_0$ voltage at a fixed confinement potential $V_{SG} = -1.0$ V. The grey circles correspond to the experimentally measured velocity at $QPC_1$, while the blue squares is the outcome of a parameter-free calculation (c.f. Supplementary Note 4). The coloured circles correspond to the velocity measured for different conductance values of $QPC_0$, i.e. $G = 2e^2/h$ (red circle), $G = 4e^2/h$ (green circle) and fully depolarised $QPC_0$ (blue circle). These data points correspond to the dotted vertical line in Fig. 2b at $V_{SG} = -1.0$ V

Our theoretical model also allows us to calculate the speed of the charge pulse assuming that only one single mode is occupied after passing the channel selection $QPC_0$ (solid red curve in Fig. 2b) and compare it to our experimental data. This mode corresponds to a single-channel Tomonaga–Luttinger plasmon (c.f. Supplementary Note 4) which is very different from the plasmon hosted by the full 28 channels. The agreement between theory and experiment is again remarkable. These observations strongly suggest that the charge pulse which is transmitted

through the lowest channel of the selection $QPC_0$ is adiabatically transferred onto the fast plasmon mode corresponding to a single-channel Tomonaga–Luttinger liquid and which we named the funneling scenario (c.f. Supplementary Note 4). We have repeated these experiments for the second quantised plateau (green data points) and find similar agreement. Hence, our data indicate that it is possible to form a very clean single channel (two-channel) Tomonaga–Luttinger liquid even though the wire contains many more active channels. We observe that the

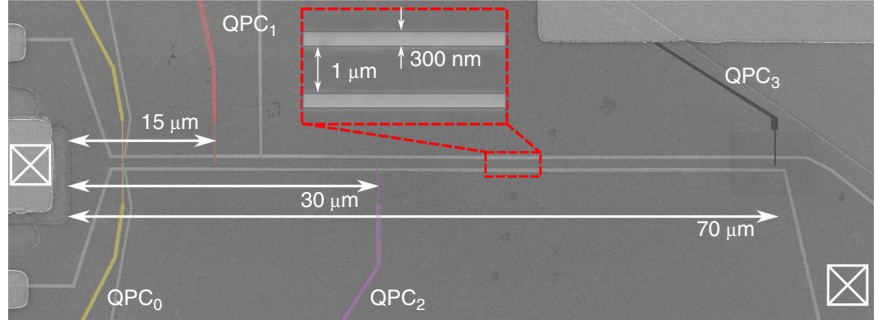

**Fig. 4** Scanning electron microscope image of the nanoelectronic device. The light grey parts correspond to the electrostatic gates. The long quasi one-dimensional channel has length of 70 μm and width of 1 μm. The two ohmic contacts, one used for the excitation of the electron pulse (left) and the other one used for current detection (right), are indicated with crossed square boxes. The three QPC switches and their respective distance from the left ohmic contact are shown with red, violet and black colours, whereas the mode selection $QPC_0$ is highlighted with yellow colour

electron pulse conserves its propagation speed for at least a distance of 25 μm (position of $QPC_2$). This is in stark contrast to experiments in the quantum Hall regime, where the wave packet fractionalises instantaneously[15]. In these experiments the electron wave packet is already fully fractionalised after a propagation distance of about 3 μm[15] with a time separation of ≈70 ps between the fast and the slow modes. In our experiment, we observe fractionalisation only at a distance well above 20 μm. At a distance of about 70 μm ($QPC_3$), we observe that the velocity is again approaching the one corresponding to the fast mode where all the channels of the quantum wire are populated. This opens the possibility to realise quantum interference experiments with single-electron pulses by only populating a single-channel plasmon mode, which has never been observed with DC measurements.

The presented time control of single-electron pulses at the picosecond level will also be important for the implementation of wave-guide architectures for flying qubits using single electrons[26]. Integrating a leviton source[27] into a wave-guide interferometer would allow to realise single-electron flying qubit architectures[26,28,29] similar to those employed in linear quantum optics[30].

Our findings also give a new insight into the recently discovered levitons[27]. As the underlying physics is independent of the actual shape of the single-electron wave packet, levitons should be regarded as a special kind of plasmon with the particularity that it does only generate electronic excitations (no holes), rather than a single-electron excitation propagating at the surface of the Fermi sea with the Fermi velocity[31].

Furthermore, our studies pave the way for studying real-time dynamics of a quantum nanoelectronic device[32] such as the measurement of the time spreading or the charge fractionalisation dynamics[10] of the electron wave packet during propagation.

## Methods

**Sample fabrication**. The sample is fabricated by depositing electrostatic gates on top of a GaAs/AlGaAs semiconductor heterostructure. The two-dimensional electron gas, which is at a depth of 140 μm, has density $n = 2.11 \times 10^{11}$ cm$^{-2}$ and mobility $\mu = 1.89 \times 10^6$ cm$^{-2}$ V$^{-1}$ s$^{-1}$, measured at 4 K. The 70 μm long electrostatic gates are defined by Ti/Au, while a Ni/Ge/Au/Ni/Au alloy is used for the ohmic contacts. A scanning electron microscope image of our sample is shown in Fig. 4.

**Time-resolved measurements of voltage pulse**. To generate a single-electron pulse, a voltage pulse with an amplitude of several tens of μV is applied to the left ohmic contact of our sample through a high bandwidth coaxial line and a 40 dB attenuation. The voltage pulses are provided by an arbitrary function generator (Tektronix AWG7122C) and have a 600 MHz repetition frequency. The generated

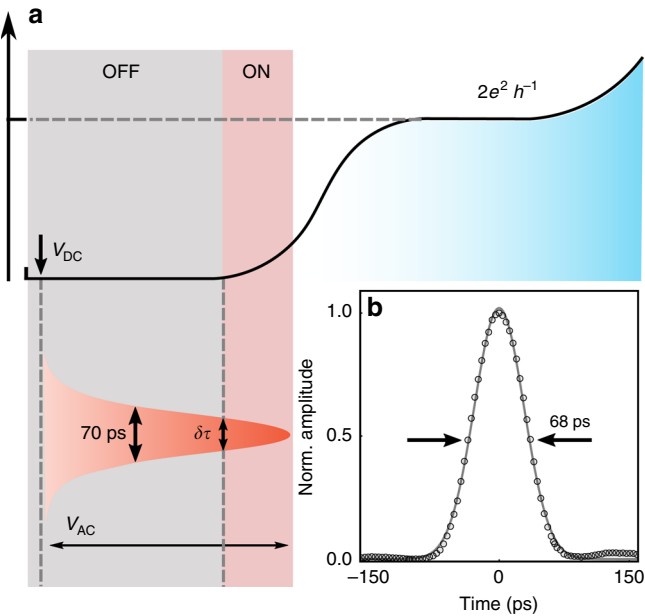

**Fig. 5** A QPC as a fast switch. **a** Operation diagram of our fast QPC switch. Initially we apply a negative DC voltage ($V_{DC}$) on the QPC to set it deep into the pinch-off regime. Then we apply a fast pulse on the QPC, $V_{AC}$, and gradually shift the DC voltage to higher values. By carefully choosing the right $V_{DC}$ we can open the QPC for a very short time, ~10 ps. **b** Time-resolved measurement of the electron pulse. The dark points are measurements and the continuous line is a Gaussian fit. The FWHM of the electron pulse is 68 ps

DC current is measured across a 10 kΩ resistor placed on the sample chip carrier at a temperature of 20 mK. The pulse train is modulated at a frequency of 12 kHz to perform lock-in measurements. A second voltage pulse is applied to one of the QPCs in order to operate it as a fast switch. By changing the time delay between generating the electron pulse and opening/closing the QPC switch we can reconstruct in a time-resolved manner the time trace of the electron pulse, following the protocol developed by Kamata et al.[19].

In order to obtain the shortest possible switching times we perform the following operations, shown in Fig. 5a. First, the QPC is set to the pinch-off regime (OFF position) by applying an appropriate negative DC voltage ($V_{DC}$). Subsequently, we apply a short voltage pulse with a fixed amplitude ($V_{AC}$) to the QPC, which allows us to open the QPC switch only for a very short time $\delta\tau$, typically below 10 ps[33]. To achieve these fast switching times we keep the $V_{AC}$ amplitude constant and we vary $V_{DC}$. As shown in Fig. 5a, when $V_{DC}$ is very negative the QPC switch remains closed for all time delays and therefore the recorded current is zero. By increasing $V_{DC}$ to the appropriate value we can open the QPC switch for a brief period of time $\delta\tau$, thus allowing us to reconstruct the

electron pulse. As the switching profile of the QPC depends on the combination of the applied DC and AC voltages as well as the very sharp conductance response we can achieve time resolutions that are shorter than those provided by our electronics. By optimising $V_{DC}$ and $V_{AC}$ amplitudes we are able to measure single-electron pulses down to a FWHM of 68 ps, as shown in Fig. 5b.

**Determination of the propagation velocity**. To determine the velocity of the electron wave packet we perfomed time-of-flight measurements for different confinement potentials (see Figs. 1, 2). For every confinement potential we carry out three independent measurements, one for each QPC (except QPC which is not connected to a bias tee). During these measurements we excite the electron wave packet and measure the time it takes to propagate to the three detection QPCs. By using the time of flight and the exact distance between the left ohmic contact (excitation location) and these three QPCs (Fig. 4) we can calculate the velocity (c.f. Supplementary Note 4).

**Data availability**. The data that support the findings of this study are available from the corresponding authors on reasonable request.

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

## Acknowledgements

We would like to dedicate this article to the late Frank Hekking who has been very implicated in the initial stage of the project. We would like to acknowledge fruitful discussions with D. Carpentier, P. Degiovanni, M. Filippone and T. Martin. S.T. acknowledges financial support from the European Union's Horizon 2020 research and innovation program under the Marie Skłodowska-Curie grant agreement No. 654603 and JSPS KAKENHI Grant Number JP18K14082. A.L. and A.D.W. gratefully acknowledge the support of DFG-TRR160, BMBF–Q.com-H 16KIS0109, and the DFH/UFA CDFA-05-06. C.B., T.M. and X.W. acknowledge financial support from the French National Agency (ANR) in the frame of its program ANR Fully Quantum Project No. ANR-16-CE30-0015-02 and QTERA No. ANR-15-CE24-0007-02. C.B. and T.M. also acknowledge the SingleEIX Project No. ANR-15-CE24-0035. X.W. and P.A. are funded by the US Office of Naval Research.

## Author contributions

G.R. and E.A. performed the experiment and analysed the data with input from S.T., G.G, M.U. and T.M., G.R. made the sample with help from S.T. and T.M., M.S. contributed to the experimental setup and timing methods. T.K., P.A. and X.W. designed the theoretical framework. P.A. performed the numerical simulations. All authors analysed the numerical data. A.L. and A.D.W. provided the high mobility hetero-structure. G.G., X.W. and C.B. wrote the paper with inputs from all authors. C.B. supervised the experimental work.
