## [Peer Review File · Nature Communications]

Reviewers' comments:

Reviewer #1 (Remarks to the Author):

The authors report on an experimental and theoretical investigation of dynamic few-electron pulses propagating through an elongated quantum conductor. Using quantum point contacts as fast switches along the conductor, the authors deduce the time-of-flight as well as the pulse shape through dc current measurements at the output. By controlling the confinement potential, the authors examine the cross-over from a multi-channel Fermi liquid to a Tomonaga-Luttinger liquid in the quasi one-dimensional limit, where the excitations become bosonic in character. As the number of channels is increased, the authors observe a clear enhancement of the pulse velocity, which is well-captured by a parameter-free theory.

I enjoyed reading the manuscript, and I find the experiment and theory beautiful and convincing. The experiment is conceptually easy to understand and the authors have struck a good balance between clarity and detail. The method section and the supplementary material contain useful additional information which supports the conclusions of the main text. The manuscript will likely influence the field of dynamic few-electron physics in quantum conductors.

I have no specific comments or concerns about the manuscript, and I believe that it can be published in Nature Communications as it is.

Perhaps as an outlook for the future, I wonder if the experiment can be adapted so that it would be possible not only to consider the individual pulses, but also to access correlations between the pulses? Personally, I would be interested in the waiting time between pulses arriving at the output. Perhaps with a clever use (or rearrangement) of the QPC switches, it would be possible to measure the distribution of waiting times?

Christian Flindt
Aalto University

Reviewer #2 (Remarks to the Author):

This paper describes time of flight measurement of short (~ 70 ps) single to few electron charge pulses injected in a quasi one-dimensional conductor. The paper investigates the renormalization of the pulse velocity caused by Coulomb interaction as a function of both the number of channels transmitted by the wire and the filtering of the transmitted pulse using a quantum point contact to select the first (or two first) QPC modes. When a single QPC mode is filtered, the authors claim that the experiment demonstrates the possibility to transmit a single mode on a distance of 20 microns.

I find the work very interesting. There is currently a strong interest in the coherent transmission of single to few electron states in a conductor. Coulomb interaction is known to strongly affect single electron states but most experiments probing these effects have been performed so far in the edge channels of the quantum Hall regime. The present experiment investigates a different situation where guided propagation is realized by the implementation of a quasi-one dimensional wire without the use of a strong magnetic field. Besides, the experiment is involved, requiring the generation and detection of short pulses at the single electron level in an elaborate nano-circuit. The experimental results are clear and compared with a theoretical model based on the bosonization framework which fully takes into account the specific geometry of the sample. The agreement with the parameter free model is excellent.

For all these reasons, I am strongly in favor of the publication of this work in Nature Communications. However, at this stage, I do not agree with the interpretation of the data provided by the authors and I do not believe that one can conclude, as the authors do, that their experiment demonstrates that mode filtering with QPC0 allows to transmit only a single mode on a distance of 20 microns, in stark contrast to experiments in the quantum Hall regime. This has to be clarified before the paper can be published.

Before I enter in the details of my understanding of the experimental results, there is an ambiguity on several occurrences in the paper in the meaning of the word "mode" (for example in the following sentence on page 9, "assuming that one single mode is occupied..." or in the expression "mode filtering"). There are two types of modes considered in the paper, the modes of the quantum wire which are the single particle eigenstates and the plasmon modes which are the eigenmodes of the velocity matrix including interactions (Eq.7 of supplementary). I think the authors should try to avoid using the word "mode" in the paper without specifying if they are talking about QPC single particle modes or plasmon (or collective) eigenmodes. My understanding of the "mode filtering" performed by QPC0 is that it selects one mode of the quantum wire (single particle mode) but does not select one collective eigenmode of the velocity matrix. As the authors do in the paper, it is very instructive to compare this experiment and its interpretation with the edge channel case and I will also rely on this comparison. In this respect, the role of the filtering QPC is similar in the edge channel case: it selects one specific single particle mode (lowest channel of the QPC) which is not an eigenmode of the velocity matrix.

My understanding of the experimental results is then the following:

Case a) QPC0 is open, there is no mode filtering (blue trace): the voltage pulse applied on the ohmic contact excites the fast plasmon mode, the pulse propagates at the fast velocity and no dispersion of the pulse is expected. The situation is similar in the edge channel case where an ohmic contact connected to two edge channels excites the fast charge mode.

Case b) QPC0 transmits one single particle mode (red trace): the single particle mode filtering initializes some specific transverse charge distribution. Computing its overlap with the eigenmodes of the velocity matrix (Eq. 7 of supplementary) one can solve the propagation of the pulse. Assuming, the most general situation where QPC0 does not initialize a single eigenmode of the velocity matrix but rather populates many eigenmodes with similar weights, I would still expect a strong reduction of the average velocity (because as shown by Figure S7.b, most of the modes propagate at the "slow velocity" and only one mode, the fast plasmon mode propagates at the fast velocity). In addition I would expect a stronger dispersion of the pulse in this case as it now decomposes on several eigenmodes with different velocities. If I am not mistaken this is precisely what is observed by the authors on Fig.3.b where the width of the red pulse seem to be approximately 30% larger than the blue one. This is also qualitatively similar to the edge channel case where the charge transmitted on the outer channel needs to be decomposed on the fast and slow plasmon eigenmodes. In this case also it is the decomposition on several collective eigenmodes that leads to a reduction of the average pulse velocity compared to the unfiltered case and to the widening of the pulse (also called the fractionalization process has the pulse eventually splits in two separate pulses).

c) By progressively increasing the number of modes transmitted by QPC0, one increases the weight of the plasmon mode (which eventually is the only one to contribute when QPC0 is fully open), the average speed then increases as the weight of the fast mode increases, and the dispersion is reduced leading to a reduced width of the pulse when QPC0 is gradually opened.

All the above interpretation agrees qualitatively (but also maybe quantitatively) with the data and additionally explains the widening of the pulse (Fig. 3.b) when mode filtering is used. However, this interpretation strongly differs from the one carried by the authors who conclude that only one mode is transmitted on a 20 microns length. Contrary to the authors, my understanding of the

data leads me to conclude that the situation is very similar to what is observed in quantum Hall edge channels: the QPC selects one single particle mode which decomposes on several collective eigenmodes resulting in a reduced average velocity and a widening of the emitted pulse. This interpretation is thus in strong disagreement with the author's conclusion on page 9: "It is therefore possible to form a very clean 1 channelwhere the electron pulse populates only the lowest mode This is in stark contrast to experiments in the quantum Hall regime where the wave packet fractionalizes already after a distance less than 3 microns". To make myself clearer, in edge channels also the electron pulse propagates "only a single mode" (outer edge channel for example) in the sense that the Coulomb interaction does not induce any scattering of electrons from one channel to the other, all the charge injected in the pulse stays in the same single particle channel. The manifestation of fractionalization comes from the widening of the electron pulse and from the emission of neutral excitations (electron hole pairs) in the other single particle channels. Fig. 3.b shows the widening of the pulse in the author's experiment, and I think this pulse widening shows that the energy carried by the pulse decreases. I think this can be attributed to the generation of electron/hole pairs in the other single particle channels in complete analogy with the edge channel case.

If I understand correctly, the interpretation of the authors relies on the agreement with the theoretical calculation "assuming that only one single mode is occupied after passing the mode selection QPC0". Here, it is crucial that the authors lift the ambiguity on which type of mode they are talking about in this sentence. Do they mean that the calculation is made assuming that QPC0 filters only one single particle mode (case 1)? Or do they mean that the calculation is made assuming that QPC0 filters only one plasmon/collective mode (case 2)?

-in case 1, I think the experiment and its agreement with the model does not prove (contrary to what the authors claim) that only one single mode is transmitted. As discussed above, the pulse transmitted by QPC0 then decomposes on several plasmon modes which is the core of the fractionalization process: single particle modes are not identical to collective eigenmodes, an electron injected in a specific single particle mode eventually decomposes on the collective modes.

-in case 2, agreement with the single plasmon mode propagation is also not enough proof for me, the experiment should demonstrate that we can rule out from the experiment the possibility that the pulse decomposes on several plasmon eigenmodes.

-the authors have means to know and provide the decomposition of the single particle mode on the collective eigenmodes by computing the overlap of the single particle mode charge density with that of the eigenmodes which velocities are plotted on Fig.S7.b. This would make the interpretation of the red trace more transparent.

-comparing the data of figure 2.b and the calculation of Figure S7.b, I am almost fully convinced that case 1 is the case considered by the authors as the measured and calculated velocity (red trace) is in between the fast and slow velocities plotted on Fig.S7.b: Red trace seem to be approximately $v \sim 3 \cdot 10^5$ (or in any case $> 2 \cdot 10^5$) for $V_{sg} = -1.6V$, but no eigenmode corresponds to this velocity on Fig.S7.b (slow eigenmodes have velocities $< 2 \cdot 10^5$, fast mode is $\sim 6 \cdot 10^5$). This shows that the average pulse velocity results from the decomposition of the pulse on several collective eigenmodes. It thus seems clear to me that the conclusion should be completely different than the one carried by the authors and qualitatively similar to what is observed in quantum Hall edge channels: after filtering one single particle mode, a pulse decomposes on several collective modes. The fractionalization can then be seen by the widening of the transmitted pulse which seems also to be seen by the authors on Fig.3.b.

I think any of these two interpretations (the one I develop in this referral and the one presented by the authors) would equally deserve publication in Nature Communication. However, the conclusion has to be supported by the data and I do not believe it is the case in the present version of the paper. The authors therefore need to either change their conclusion/interpretation or convince me that their interpretation is correct. In addition, I think that the possibility to tune the decomposition on the collective modes and the velocities of the collective modes by changing VSG is very interesting and could provide quantitative differences with the edge channel even though the underlying physics demonstrated in this experiment is, for me, very similar.

I have finally a comment/suggestion regarding the emphasis put by the authors in their title and abstract on the single electron nature of the generated pulses. I think it is misleading: nothing in the present experiment depends on the number of charges carried by the pulses, everything is linear in the pulse amplitude and no dependence on the charge is expected, which is what is observed by the authors. I would recommend to reduce the emphasis on the single electron nature of the pulses in the title and abstract: for example a sentence such as “even in the quantum regime where the pulses carry one or less electrons” is completely misleading as it suggests that the naïve intuition would lead one to think that interaction effects should be different for small pulses amplitudes. On the contrary, the naïve intuition suggests that the propagation of the bosonic collective modes does not depend on their amplitude. I would however keep the discussion on single electron pulses in the introduction and conclusion as a motivation of the work.

-final suggestion, the authors should mention in the caption of Figure 2 where the speed is measured (QPC1, 2 or 3?).

Sincerely,
Gwendal Fève
Professor at University Pierre et Marie Curie
Laboratoire Pierre Aigrain

Reviewer #1

We would like to thank the referee for his report and that he finds our article suitable for publication in Nature Communication.

Concerning the referee's remark on the waiting time distribution:

“Perhaps as an outlook for the future, I wonder if the experiment can be adapted so that it would be possible not only to consider the individual pulses, but also to access correlations between the pulses? Personally, I would be interested in the waiting time between pulses arriving at the output. Perhaps with a clever use (or rearrangement) of the QPC switches, it would be possible to measure the distribution of waiting times?”

This is indeed an interesting point.

One could indeed think of placing a quantum dot within the entrance of the quantum wire and measure the resulting arrival times of the emitted electron wave packet for different tunnel barrier values when triggered with a periodic voltage pulse.

To be able to measure the waiting time distribution, however, one would have to implement single-shot detection of the arriving single-electron wave packet.

We are presently developing such a single-shot detector based on a charge qubit and hopefully in the near future we will be able to measure the waiting time distribution.

Reviewer #2

We would like to thank the reviewer for the careful and very critical reading of our manuscript. The referee appreciates very much our work and *“is strongly in favor of the publication of our work in Nature Communications”*. The referee, however, does not agree with *“the interpretation of the data provided by us and does not believe that one can conclude that our experiment demonstrates that mode filtering with QPC0 allows to transmit only a single mode on a distance of 20 microns, in stark contrast to experiments in the quantum Hall regime.”*

The referee asked us to clarify this issue before the paper can be published.

Before answering in detail the comments of the referee, let us emphasize that the main outcome of our work is the understanding of the propagation of an ultrashort charge pulse in a multi-channel quantum wire when tuning the system from the 1D limit towards the 2D limit. Our experiment shows for the first time how the speed of the bosonic plasmon mode evolves when going from a Tomonaga-Luttinger liquid towards a 2D Fermi liquid.

In addition we provide a parameter free theory which reproduces very well our experimental data.

Only the *interpretation* of the experimental data of the second part of our manuscript where we use the “mode filtering” QPC is questioned by the referee.

Our referee proposes an alternative interpretation of the “mode filtering” QPC effect. After an extensive email and phone discussion with our referee, we conclude that his interpretation - which appears to be valid in his own experiments at high magnetic field - is not supported by

our experimental & theoretical work at zero magnetic field. Our arguments, which are detailed below and in a new section of the supplementary material (with a new figure) are summarized as follows:

- First and above, our referee interpretation would imply that our measured signal should split into two well separated peaks for measurement at QPC 2 and QPC 3. We did not observe any such “fractionalization”.
- Second, our interpretation perfectly matches our theoretical treatment in the absence of any adjustable parameter.
- Third, there exists a characteristic relaxation time scale that separates the two regimes described by the two interpretations. This time scale has neither been measured nor predicted. We see no particular reason to favor *a priori* one scenario over another, and our experimental data strongly favors our hypothesis.
- Fourth, our referee interpretation would imply that the velocity would depend on the confining potential. We do not observe this voltage dependence.

In the following we give a very detailed response to the points raised by the referee. In addition we provide further theoretical data to support the interpretation of our experimental data.

Comment of Referee #2:

(1)..., there is an ambiguity on several occurrences in the paper in the meaning of the word “mode” (for example in the following sentence on page 9, “assuming that one single mode is occupied...” or in the expression “mode filtering”). There are two types of modes considered in the paper, the modes of the quantum wire which are the single particle eigenstates and the plasmon modes which are the eigenmodes of the velocity matrix including interactions (Eq.7 of supplementary). I think the authors should try to avoid using the word “mode” in the paper without specifying if they are talking about QPC single particle modes or plasmon (or collective) eigenmodes.

We apologize for this ambiguity. In the revised manuscript, we have eliminated this ambiguity by explicitly stating (at the beginning of the paragraph “Effect of Coulomb interaction on the propagation velocity”):

To distinguish between single-particle states and collective modes, we will use throughout the manuscript the term $\textit{channel}$ whenever referring to single-particle states and \textit{mode} when referring to collective modes.

Comment of Referee #2:

(2) My understanding of the “mode filtering” performed by QPC0 is that it selects one mode of the quantum wire (single particle mode) but does not select one collective eigenmode of the velocity matrix. As the authors do in the paper, it is very instructive to compare this experiment and its interpretation with the edge channel case and I will also rely on this comparison. In this respect, the role of the filtering QPC is similar in the edge channel case: it selects one specific single particle mode (lowest channel of the QPC) which is not an eigenmode of the velocity matrix.

We do agree with the referee that the QPC selects a single-particle state and NOT an eigenmode of the velocity matrix.

We have emphasized this point in the revised version of the manuscript. At the same time, we find that this filtering *modifies* the plasmon velocity. This point was perhaps a bit ambiguous before and is now explained in detail in the new section of the supplementary material.

Comment of Referee #2:

(3) The manifestation of fractionalization comes from the widening of the electron pulse and from the emission of neutral excitations (electron hole pairs) in the other single particle channels. Fig. 3.b shows the widening of the pulse in the author's experiment, and I think this pulse widening shows that the energy carried by the pulse decreases. I think this can be attributed to the generation of electron/hole pairs in the other single particle channels in complete analogy with the edge channel case.

We agree that a pulse widening could - in principle - occur due to the fractionalization process if for instance 2 charge pulses of approximately equal weight (e.g. in the quantum Hall effect for $\nu=2$) are separated due to different propagation velocities.

The observed widening of the pulse seen in Fig. 3.b, however, cannot be explained by the fractionalization process proposed by the referee. To demonstrate this we have simulated the pulse shape by supposing such fractionalization process using the velocity matrix (eq. 7 of suppl. material). If the widening of the pulse is due to a superposition of different modes travelling at different velocities, we should see the fractionalization at QPC2 and QPC3 as shown in the simulation presented in the additional section of the revised supplementary materials (see figure S8)

In our experiment we did not see such a fractionalization.

The observed widening of the pulse width is probably coming from other physical origins. When the pulse enters the region between OPC0 and QPC1 we effectively create a Fabry-Perot cavity. The pulse bouncing back and forth within this cavity affects the measured pulse shape and leads to a widening and slightly asymmetric shape. This is seen very clearly when making a cavity with QPC0 and QPC3 where the pulse width is shorter than the cavity length (data not shown in the manuscript). There might be other effects such as inelastic scattering which will induce a broadening of the pulse for propagation distances longer than the mean free path which is of the order of 10 micrometers in our sample.

A systematic study of the pulse widening as a function of propagation length could certainly be done, but is beyond the scope of the present manuscript. Let us mention, however, that such a study is not simple, as the detected pulse width depends on the individual QPC operation as a fast on/off switch. The measured pulse width is a convolution of the voltage pulse generated at the ohmic contact and the voltage pulse applied to the QPC. As explained in the methods section (Figure 5) the QPC switch is opened only during a fraction of this applied voltage pulse. The opening-time, however, depends on the actual shape of the pinch-off curve of the detection QPC close to the pinch-off. As the pinch-off curves are not identical between the different QPCs (QPC1, QPC2, QPC3) we observe differences in the pulse width. This effect is actually seen in figure 1b). The fast plasmon detected at QPC2 has a $\sim 10\%$ larger width than when detected at QPC 1 or QPC 3.

Comment of Referee #2:

(4) If I understand correctly, the interpretation of the authors relies on the agreement with the theoretical calculation “assuming that only one single mode is occupied after passing the mode selection QPC0”. Here, it is crucial that the authors lift the ambiguity on which type of mode they are talking about in this sentence.

We have eliminated this ambiguity in the revised manuscript.
See our response to comment (1).

(5) Do they mean that the calculation is made assuming that QPC0 filters only one single particle mode (case 1)? Or do they mean that the calculation is made assuming that QPC0 filters only one plasmon/collective mode (case 2)?

We are sorry that this point was not clear in the previous version of our manuscript. As mentioned above, we have now added a section in the supplementary material where we explain this issue in detail.

In short, the QPC₀ selects a single particle state and it is then transformed it into a single-channel plasmon mode (For details see suppl. materials).

How this process is taking place on the microscopic level is at present not clear. However, our experiments give very valuable insight into this interesting problem. Our experiment shows that the resulting charge pulse (after selection QPC₀) propagates with a speed much slower than the speed of the collective fast plasmon mode.

We find a speed that is consistent with the speed of a plasmon mode assuming that all the charge distribution is propagated by a single-channel plasmon mode (red curve in figure 2b) or by a two-channel plasmon mode (green curve in figure 2b).

The calculated speeds for these 2 specific configurations corresponds to a truncated velocity matrix of 1x1 (red curve) and 2x2 (green curve). Please see the supplementary material (end of section IV-B and new section IV-C) for more details.

These experimental findings combined with our parameter free theory strongly suggests that only a single (two) collective mode(s) is (are) transmitted after the filtering QPC₀ over a distance up to approximately 25 micrometers.

In addition, the fact that the speed is independent on the confinement potential (the number of channel N within the quantum wire) corroborates this picture.

If the scenario pointed out by the referee would apply, one would expect that the propagation speed should depend on N.

Comment of Referee #2:

(6) -in case 1, I think the experiment and its agreement with the model does not prove (contrary to what the authors claim) that only one single mode is transmitted. As discussed above, the pulse transmitted by QPC0 then decomposes on several plasmon modes which is the core of the fractionalization process: single particle modes are not identical to

collective eigenmodes, an electron injected in a specific single particle mode eventually decomposes on the collective modes.

We do agree with the referee that a specific single-particle mode eventually decomposes on the collective eigenmodes. This is indeed what we observe on long time scales, i.e. after a propagation distance of 65 micrometers, where we find that the propagation speed of the charge pulse is again close the fast plasmon mode for the wide quantum wire containing N channels (without filtering QPC).

On a short time scale, however, our experimental data suggests that the single-particle state is projected onto a single collective eigenmode of the quantum wire.

To understand this fractionalization process on a microscopic level is of course an interesting, but challenging question and beyond the scope of this paper.

We refer to this open question in the conclusion sentence and mention it as interesting outlook.

Comment of Referee #2:

(7) -in case 2, agreement with the single plasmon mode propagation is also not enough proof for me, the experiment should demonstrate that we can rule out from the experiment the possibility that the pulse decomposes on several plasmon eigenmodes.

-the authors have means to know and provide the decomposition of the single particle mode on the collective eigenmodes by computing the overlap of the single particle mode charge density with that of the eigenmodes which velocities are plotted on Fig.S7.b. This would made the interpretation of the red trace more transparent.

As mentioned above (3), we have calculated the decomposition of the single-particle mode on the collective eigenmodes by computing the overlap of the single particle mode charge density with that of the eigenmodes of the velocity matrix.

Our calculations show that our experimental findings are inconsistent with the model proposed by the referee (for details see response to comment (3)).

As mentioned above (5), we find that the speed of the propagating charge pulse is in excellent agreement with the speed of a plasmon mode where all the charge distribution is carried by a single plasmon mode (e.g. red curve in figure 2b).

Comment of Referee #2:

8) I think any of these two interpretations (the one I develop in this referral and the one presented by the authors) would equally deserve publication in Nature Communication. However, the conclusion has to be supported by the data and I do not believe it is the case in the present version of the paper. The authors therefore need to either change their conclusion/interpretation or convince me that their interpretation is correct. In addition, I think that the possibility to tune the decomposition on the collective modes and the velocities of the collective modes by changing VSG is very interesting and could provide quantitative differences with the edge channel even though the underlying physics demonstrated in this experiment is, for me, very similar.

We agree that the physics is certainly similar to the one in the quantum Hall effect. We believe however that the microscopic mechanism as well as the length/time scales on which the fractionalization of the charge pulse occurs are very much different.

In the quantum Hall regime the fractionalization process has to appear almost instantaneously after injecting the charge into the outer edge channel, otherwise the separation between the neutral and fast mode would not be measurable after a short propagation length of $\sim 3 \mu\text{m}$ (see Freulon et al. Nature Communication 7854, 2015).

In our case we do not observe this fractionalization up to a length scale of at least 25 micrometers, which corresponds to a timescale of $\sim 80 \text{ ps}$ (taking the propagation speed of $\sim 300 \text{ km/s}$; red curve in fig. 2b).

These differences may come from the fact that the QPC filtering works differently than the electron injection into an edge state in the QHE. In addition, in our system the channels are not coupled capacitively (wave function overlap between geometrically separated channel as in the QHE) but due to a wave function overlap between channels separated in energy.

Comment of Referee #2:

(9) I have finally a comment/suggestion regarding the emphasis put by the authors in their title and abstract on the single electron nature of the generated pulses. I think it is misleading: nothing in the present experiment depends on the number of charges carried by the pulses, everything is linear in the pulse amplitude and no dependence on the charge is expected, which is what is observed by the authors. I would recommend to reduce the emphasis on the single electron nature of the pulses in the title and abstract: for example a sentence such as “even in the quantum regime where the pulses carry one or less electrons” is completely misleading as it suggests that the naïve intuition would lead one to think that interaction effects should be different for small pulses amplitudes. On the contrary, the naïve intuition suggests that the propagation of the bosonic collective modes does not depend on their amplitude. I would however keep the discussion on single electron pulses in the introduction and conclusion as a motivation of the work.

We agree with the referee that the physics we discuss in our manuscript does neither depend on the amplitude nor on the shape of the voltage pulse.

Following the suggestion of the referee, we have changed the title to:

“Unveiling the bosonic nature in an ultrashort few-electron pulse”

- We also have rephrased the abstract a little in order to take into account the suggestions of the referee.

Comment of Referee #2:

(10) final suggestion, the authors should mention in the caption of Figure 2 where the speed is measured (QPC1, 2 or 3?).

- the speed was measured at QPC 3; we have added this information in the caption of Fig. 2.

List of major modifications:

- Title has been slightly changed as requested by the referee.
- The “abstract” has been slightly modified as requested by the referee.
- We have added an additional section into the supplementary which explains in detail how we have obtained the red and green curves shown in figure 2 of the main text.
- We have added an additional figure in the supplementary materials (figure S8).
- We have added a reference which was not available when submitting the 1st version of the manuscript [Ref 26] Bäuerle et al. Rep. Prog. of Phys. 81 056503 (2018)

REVIEWERS' COMMENTS:

Reviewer #1 (Remarks to the Author):

Based on the response by the authors and the revisions, I believe that the manuscript can now be published.

Christian Flindt

Reviewer #2 (Remarks to the Author):

In am now convinced by the authors interpretation of their data. In particular, the detailed discussion in the supplementary material of the differences between the funneling scenario and the filtering scenario unambiguously demonstrates that the experimental data is only consistent with the funneling case. Additionally, the ambiguity between the single particle channels and the collective modes has been lifted in the revised version. I thus recommend the publication in Nature Communications.